# Investigation of Metastable Low Dimensional Halometallates

**DOI:** 10.3390/molecules27010280

**Published:** 2022-01-03

**Authors:** Navindra Keerthisinghe, Matthew S. Christian, Anna A. Berseneva, Gregory Morrison, Vladislav V. Klepov, Mark D. Smith, Hans-Conrad zur Loye

**Affiliations:** Department of Chemistry and Biochemistry, University of South Carolina, Columbia, SC 29208, USA; KEERTHID@email.sc.edu (N.K.); MCHRISTI@cec.sc.edu (M.S.C.); bersenea@email.sc.edu (A.A.B.); MORRI383@mailbox.sc.edu (G.M.); klepov@northwestern.edu (V.V.K.); MDSMITH3@mailbox.sc.edu (M.D.S.)

**Keywords:** halometallate, hydrothermal synthesis, dimensional reduction

## Abstract

The solvothermal synthesis, structure determination and optical characterization of five new metastable halometallate compounds, [1,10-phenH][Pb_3.5_I_8_] (**1**), [1,10-phenH_2_][Pb_5_I_12_]·(H_2_O) (**2**), [1,10-phen][Pb_2_I_4_] (**3**), [1,10-phen]_2_[Pb_5_Br_10_] (**4**) and [1,10-phenH][SbI_4_]·(H_2_O) (**5**), are reported. The materials exhibit rich structural diversity and exhibit structural dimensionalities that include 1D chains, 2D sheets and 3D frameworks. The optical spectra of these materials are consistent with bandgaps ranging from 2.70 to 3.44 eV. We show that the optical behavior depends on the structural dimensionality of the reported materials, which are potential candidates for semiconductor applications.

## 1. Introduction

In recent years, hybrid organic-inorganic perovskites (HOIPs) have gained attention due to their interesting optical and electronic properties, particularly for their potential use as photovoltaic materials [1,2,3,4,5,6]. This application is further favored by their amenability to solution processing. Many of the phases that are studied specifically for photovoltaic applications crystallize in perovskite and perovskite-related structures. The simple perovskite structure, with the general formula ABX_3_, consists of corner-sharing BX_6_ octahedra with the A cation located in the center of a cube made up of eight corner-sharing octahedra. In HOIPs, the typically inorganic A cation is replaced by an organic cation, which can impact the dimensionality of the HOIPs and typically depends on the size of the specific organic cation used. In general, larger organic cations lead to layered structures, such as the Ruddlesden–Popper phases with the general formula A_2_BX_4_ [6,7] that consist of sheets of corner-shared BX_6_ octahedra that are separated from each other by interlocking organic cations. Compositionally related to HOIPs are the classes of low dimensional organic-inorganic halometallates that exhibit a vast structural diversity. Many of these materials are what we would describe as “metastable”, as their temperature stability is quite limited.

Main group halometallate compounds (HMCs) constitute a class of hybrid materials that have been extensively studied for their high structural diversity and tunable optical properties [8,9,10,11,12,13,14,15,16,17]. These materials often consist of an inorganic halometallate anion and a charge-balancing organic cation. Their syntheses are typically based on crystallization out of concentrated hydrohalic acid solutions, and products are often obtained as single crystals that form during the slow cooling of the solution containing the organic species [18]. These phases are also what we would describe as “metastable”.

The anion, composed of MX_n_ polyhedra (M = main group metal, X = halide, n = integer), can be connected via corner-, edge- or face-sharing arrangements to produce 0D, 1D, 2D or 3D structures [19]. For example, iodoplumbates, which were extensively studied by Krautscheid et al., exhibit a wide range of anionic polyhedra, including [Pb_2_I_6_]^2−^, [Pb_3_I_8_]^2−^, [Pb_3_I_10_]^4−^, [Pb_7_I_18_]^4−^ and [Pb_10_I_28_]^8−^ [13,20]. The majority of these HMCs have organic monoamines/polyamines, N-containing organic ligands or metal coordination complexes as the charge balancing cationic species. This leads to an extremely large number of potential compositions of novel HMCs that can be crystallized in different structures and different dimensionalities, leading to the ability to compositionally adjust the optical properties. To date, many of the reported HMCs have exhibited semiconductor behavior [13,14,21,22]. It needs to be stressed that for these materials, the specific products and product structures that form are extremely sensitive to the reaction conditions used, signaling their metastable nature.

Herein, we report the solvothermal synthesis of five new HMCs that vary in dimensionality from 1D to 3D. In Figure 1, we report on three different iodoplumbates, [1,10-phenH][Pb_3.5_I_8_] (**1**), [1,10-phenH_2_][Pb_5_I_12_]·(H_2_O) (**2**) and [1,10-phen][Pb_2_I_4_] (**3**), that exist in a narrow phase space. The change of the halide from iodide to bromide results in the structurally different [1,10-phen]_2_[Pb_5_Br_10_] (**4**), and the cation change from Pb^2+^ to Sb^3+^ results in the 1D chain structured [1,10-phenH][SbI_4_]·(H_2_O) (**5**). We observed that minute changes in the synthesis conditions were sufficient to significantly affect the compositions and structures of the crystallized species. We were able to use low reaction temperatures to obtain the reported metastable materials as plentiful crystals. In comparison to the three-dimensional precursor halides, these materials exhibit dimensional reduction in their crystal structures that induces changes in their optical properties.

## 2. Results and Discussion

### 2.1. Synthesis

There are many ways to think about metastable materials and exactly what that term implies about the nature of a material, its composition, its structure and its stability. One way is to consider metastable materials as structures and compositions that form in a local energy minimum and that are not the most thermodynamically stable phase that exists at the global energy minimum [23,24,25,26,27,28,29,30,31]. Synthesizing such metastable materials therefore requires reaction conditions that kinetically result in the formation of a metastable phase and that are mild enough, often by being carried out at low temperatures, to avoid sliding into the global energy minimum. In this context, metastable- and kinetic-phase are often used interchangeably. Generally, solid state reactions, carried out at high temperatures, result in the most thermodynamically stable phases and, therefore, are not typically used to target metastable structures and compositions. That is better done via low temperature solution routes, such as low melting fluxes or mild hydrothermal conditions, that operate under conditions in which a metastable material will persist to be isolated rather than thermally convert to a more stable phase.

The halometallate materials discussed in this paper were all synthesized under extremely mild conditions where numerous local minima appear to exist side by side and where slight perturbations of the synthetic conditions resulted in the stabilization and subsequent isolation of one or another structure and composition. Surprisingly, very slight changes in temperature, pH and concentration of the reaction mixture resulted in different products with significantly different crystal structures (Figure 2). Products **1**, **2** and **3** form in only a very narrow synthesis space, and depending on the precise synthesis conditions, either one or several of the three phases would form. When Br^−^ was used instead of I^−^, no analogues of **1**, **2** or **3** were observed, with product **4** forming instead. The formation of product **4** competed with the formation of an unidentified amorphous product, the outcome being strictly dependent on temperature and reactant concentration. The latter had the greatest impact, and **4** only formed when using a 10-fold reduced concentration versus the one that resulted in the formation of **1**. These mild conditions are convenient and allowed us to modify the reaction conditions sufficiently to access the metastable materials reported herein.

#### 2.1.1. [1,10-phenH][Pb_3.5_I_8_] and [1,10-phenH_2_][Pb_5_I_12_]·(H_2_O)

The solvothermal reaction between PbI_2_, 1,10-phenanthroline and HI resulted in yellow needle-like crystals of [1,10-phenH][Pb_3.5_I_8_] (**1**). The reactions were heated to 160 °C for 12 h in a 12 mL glass pressure vessel that was sealed with a threaded Teflon plug with additional Teflon tape added to prevent solvent loss. When solvent loss did occur when not using the tape, the reaction produced a combination of orange plate-like crystals of [1,10-phenH_2_][Pb_5_I_12_]·(H_2_O) (**2**) on the tube wall and red diamond shaped crystals at the bottom of the tube (See Appendix A). The red crystals diffracted poorly, and their structure could not be solved. The orange crystals, on the other hand, were of fine quality and allowed the structure of **2**, containing the doubly protonated phenanthrolinium cation [1,10-phenH_2_]^2+^, to be determined. We infer that upon evaporation of the solvent, the concentration and/or acidity of the reaction media increased and resulted in the latter products. However, attempts of synthesizing phase pure **2** or only red diamond shaped crystals were unsuccessful and resulted in material **1** in many attempts.

#### 2.1.2. [1,10-phen][Pb_2_I_4_]

In attempts of purifying **2**, the use of lower temperatures was investigated. However, when the temperature was reduced by 10 °C, from 160 to 150 °C, the reaction yielded a mixture of yellow plate-like crystals of [1,10-phen][Pb_2_I_4_] (**3**) and **1**. When the reaction was repeated, starting with stoichiometric ratios of PbI_2_ and 1,10-phenanthroline to match the composition of **3**, only an amorphous product resulted. Due to the difference in crystal morphology, we were able to handpick and isolate crystals of **3** under the microscope for characterization.

#### 2.1.3. [1,10-phen]_2_[Pb_5_Br_10_]

Upon changing the halide from iodide to bromide, pink plate-like crystals of [1,10-phen]_2_[Pb_5_Br_10_] (**4**) were obtained. When using the exact molar amounts of 0.30 mmol of PbBr_2_ and 0.10 mmol of 1,10-phenanthroline as in synthesis 1, the reaction yielded an excessive amount of an amorphous product that looked like white cotton wool when in solution. After numerous attempts of reducing the concentration, the optimal molar amounts were found to be 0.03 mmol of PbBr_2_ and 0.01 mmol of 1,10-phenanthroline. It was also noticed that the crystallinity of the material was greatly impacted by the reaction temperature and cooling rate, and only when the reaction vessels were removed from the oven as soon as it reached room temperature, immediately placed into a cooler environment (<20 °C) and allowed to sit undisturbed were plentiful crystals of **4** obtained.

#### 2.1.4. [1,10-phenH][SbI_4_]·(H_2_O)

The use of antimony instead of lead, under the same reaction conditions used in reaction **1**, resulted in the formation of yellow plate-like crystals of [1,10-phenH][SbI_4_]·(H_2_O) (**5**). Each reaction produced an average yield of ~45 mg of product (~18% yield with respect to Sb^3+^).

### 2.2. Crystal Structure Solutions

#### 2.2.1. [1,10-phenH][Pb_3.5_I_8_]

The compound crystallizes in the monoclinic system, space group *C2/m* (No. 12). During the initial structure solution by SHELXT, two Pb and three I positions were found, resulting in layers with a nominal composition of PbI_2_. The residual electron density peaks in between the layers corresponded to aromatic rings of a phenanthroline molecule, which is disordered over two positions by a two-fold rotational axis, giving a composition of (1,10-phen)Pb_4_I_8_. (Appendix A). One of the two Pb sites, Pb1, had a notably larger thermal displacement ellipsoid than the other site and was freely refined to 0.75 occupancy. Given that free refinement of the other heavy element sites did not result in any significant deviation from unit occupancy, the Pb1 site was fixed at 0.75 occupancy, resulting in a non-charge-balanced formula (phen)Pb_3.5_I_8_. The lack of a positive charge in the composition is likely compensated by the protonation of the phenanthroline molecules, which cannot be unambiguously confirmed by single crystal X-ray diffraction but results in a charge-balanced formula (phenH)Pb_3.5_I_8_. Although a reasonable structural model was obtained, a high R_1_ value of 0.0896 and high positive residual electron density of 9.2 e/Å^3^ indicated the presence of unaccounted twinning. The twin law of (1 0 1.456 0 –1 0 0 0 –1) was found using TwinRotMat program of the PLATON package and introduced to the model, resulting in a BASF value of 0.25401 and relatively low R_1_ and residual electron density of 0.0427 and 3.80 e^−^/Å^3^, respectively.

#### 2.2.2. [1,10-phenH_2_][Pb_5_I_12_]·(H_2_O)

The compound crystalizes in the orthorhombic system, space group *Cmme* (No. 67). The initial structure solution by SHELXT resulted in three Pb and five I sites, forming a 3D hollow structure with a nominal composition of Pb_5_I_12_. The residual densities in between the channels were assigned to di-protonated phenanthrolinium cation ([1,10-phenH_2_]^2+^) and one water molecule. The asymmetric unit consists of one [Pb_3_I_5_] unit, a quarter of a disordered phenanthrolinium cation and a half-occupied water molecule (Appendix A). The phenanthrolinium cation is disordered over two orientations related by a two-fold rotational axis throughout the crystal, resulting in the superposition of the two orientations. The C2/N2 were refined with identical coordinates and displacement parameters. The water (O1W) is disordered over the two orientations of the phenanthrolinium cation. All non-hydrogen atoms were refined with anisotropic displacement parameters except O1W. The hydrogen atoms bonded to carbons were fixed in geometrically idealized positions and included as riding atoms with *d*(C–H) = 0.93 Å and *U*iso(H) = 1.2*U*eqI. To maintain charge balance, the 1,10-phenanthroline molecule must be doubly protonated. The nitrogen bound hydrogen could not be located and was arbitrarily placed on nitrogen N2 with *d*(N–H) = 0.853 Å and *U*iso(H) = 1.2*U*eq(N). The hydrogen atoms on the water molecule (O1W) could not be located. The largest residual electron density peak in the final difference map is 0.84 e^−^/Å^3^, located 0.636 Å from O1W.

#### 2.2.3. [1,10-phen][Pb_2_I_4_]

The compound crystalizes in the monoclinic system, space group *I2/a* (No. 15). The initial structure solution by SHELXT resulted in two Pb, four I and eight C positions giving a double chain structure in space group *P2_1_/c* (Appendix A). The residual electron density peaks in between the chains were attributed to aromatic rings of 1,10-phenanothroline molecules. The structure was checked for missing symmetry with the Addsym program implemented in PLATON software, and higher symmetry (space group *I2/a)* was found. The asymmetric unit consists of two Pb atoms sharing two I atoms, and Pb1 is connected to half of a phenanthroline molecule via a Pb1-N bond. The distance between C6-C6a atoms was restricted to 1.419 (0.02) Å to maintain the aromatic ring structure in phenanthroline. All non-hydrogen atoms were refined with anisotropic displacement parameters, and the hydrogen atoms bonded to carbons were fixed in geometrically idealized positions and included as riding atoms with *d*(C–H) = 0.93 Å and *U*iso(H) = 1.2*U*eq(C). The largest residual electron density peak in the final difference map is 1.39 e^−^/Å^3^, located 1.475 Å from C5.

#### 2.2.4. [1,10-phen]_2_[Pb_5_Br_10_]

The compound crystalizes in the monoclinic system, space group *P2_1_/c* (No. 14). The asymmetric unit consists of a 1,10-phenanthroline molecule connected to a Pb_3_Br_5_ unit resulting in a layered structure (Appendix A). There are two distinct Pb octahedral sites. Pb1 is in a distorted octahedra connected to the 1,10-phenanthroline via Pb-N bonds and 4 bromine atoms, while the other is a PbBr_6_ unit. All non-hydrogen atoms were refined with anisotropic displacement parameters, and the hydrogen atoms bonded to carbons were fixed in geometrically idealized positions and included as riding atoms with *d*(C–H) = 0.93 Å and *U*iso(H) = 1.2*U*eq©. The largest residual electron density peak in the final difference map is 1.31 e/Å^–3^, located 0.928 Å from Pb2.

#### 2.2.5. [1,10-phenH][SbI_4_]·(H_2_O)

The compound crystallizes in the triclinic system. Because of crystallographic disorder observed in the structure (Appendix A), models were developed in both space groups *P*1 (No. 1) and *P*-1 (No. 2); the centrosymmetric group *P*-1 was ultimately determined to be correct. Refinement in *P*1 (No. 1) showed the same disorder observed in *P*-1 and was unstable. Examination of precession images synthesized from the dataset did not reveal any signs of a different (larger) unit cell which might resolve the observed disorder. The asymmetric unit in *P*-1 consists of one SbI_4_ unit and a disordered complex consisting of one phenanthrolinium cation and one water molecule. The phenH/H_2_O complex occupies two fractionally occupied orientations, with the disorder taking the form of a (non-crystallographic) 180° rotation about a line passing through atoms C3–C1–C12–C10. The two phenanthrolinium disorder components (A/B) are coplanar, and most atoms are superimposed on those of the other component, the exceptions to this being atoms C6(A/B) and C7(A/B) and the water molecules O1(A/B) situated near the nitrogen atoms. Coordinates and displacement parameters for superimposed atoms were held equal. The disorder fractions are therefore determined primarily by atoms C6, C7 and O1, refined to A/B = 0.666(4)/0.334(4) (Appendix A). All non-hydrogen atoms were refined with anisotropic displacement parameters except the water oxygen atoms (isotropic). Hydrogen atoms bonded to carbon were placed in geometrically idealized positions and included as riding atoms with *d*(C–H) = 0.95 Å and *U*iso(H) = 1.2*U*eq(C). It was not possible to locate the hydrogen atoms on the phenanthrolinium nitrogen atoms which must be present for charge balance; they were therefore semi-arbitrarily placed on the nitrogen located closest to water (N2A and N2B) with *d*(N–H) = 0.86 Å and *U*iso(H) = 1.2*U*eq(N), under the assumption of a NH–O hydrogen bond to the nearby water oxygen. Hydrogen atoms could neither be located nor were they calculated for the water molecules. The largest residual electron density peak in the final difference map is 0.56 e^−^/Å^3^, located 0.75 Å from I4.

### 2.3. Crystal Structure Description

#### 2.3.1. [1,10-phenH][Pb_3.5_I_8_]

Compound **1** crystallizes in the monoclinic space group *C2/m* (No. 12). The material forms as a 2D structure containing [Pb_3.5_I_8_]^−^ layers separated by monoprotonated 1,10-phenanthrolinium cations ([1,10-phenH]^+^) (Figure 3). Interestingly, the structure of the lead-deficient [Pb_3.5_I_8_]^−^ layer is substantially different from the layers constituting the hexagonal PbI_2_ precursor used in the synthesis. While the hexagonal PbI_2_ structure is based on flat layers containing a single crystallographic octahedral site, the lead iodide layers of **1** are corrugated (Figure 3) and contain two distinct octahedral sites occupied by Pb(1) and Pb(2). The Pb(1) site is only 75% occupied, while the Pb(2) site is fully occupied, resulting in the Pb_3.5_I_8_ layer composition. When viewed down the *c*-axis, the negatively charged layer exhibits a stepwise arrangement of Pb(1)I_6_ and Pb(2)I_6_ edge-shared octahedra (Figure 3). Both octahedra are slightly distorted towards the [1,10-phenH]^+^ cations and have average bond lengths of 3.215 Å and 3.2316 Å for Pb(1)-I and Pb(2)-I, respectively. The [1,10-phenH]^+^ cations are disordered over two positions by a two-fold rotational axis, and all atoms are superimposed except for C(5). The C(1) position is partially occupied by both C and N. The presence of the proton attached to the nitrogen was confirmed by infrared (IR) spectroscopy; however, it was not possible to locate the proton when solving the structure.

The presence of the [1,10-phenH]^+^ cations disrupts the stacking of the PbI_2_ layers and forces a shift between them which generates a long, a medium and a short I-I separation (Appendix A). The interlayer I-I distance observed in hexagonal PbI_2_ is 4.663 Å, noticeably longer than the short I-I distance of 4.233 Å between the Pb_3.5_I_8_ layers found in **1**; the long separation is 12.910 Å. The stacking arrangement in **1** creates this short distance, undoubtedly aided by the coulombic attraction between the anionic [Pb_3.5_I_8_]^−^ layers and the intervening [1,10-phenH]^+^ cations. This change of the stacking arrangement also results in a subtle change in the bandgap for **1**, compared to PbI_2_, as discussed later.

#### 2.3.2. [1,10-phenH_2_][Pb_5_I_12_]·(H_2_O)

Compound **2** crystalizes in the orthorhombic space group *Cmme* (No. 67). This material precipitates upon solvent loss when **1** would otherwise be obtained. It consists of a 3D channel structure, where disordered [1,10-phenH_2_]^2+^/H_2_O complexes reside in the channels (Figure 4). There are three distinct Pb sites, and the 3D structure contains two different PbI_6_ octahedral chains. Chain 1 contains Pb(1)I_6_ octahedra that face share with Pb(2)I_6_ octahedra to form trimers; the trimers connect to each other via edge-sharing to form chain 1 that has a zig zag shape. In contrast, the linear chain 2 consists of all edge-sharing Pb(3)I_6_ octahedra. The two chains are oriented perpendicular to each other and connect by sharing edges between 2 Pb(3)I_6_ and Pb(1)I_6_ of one chain 1 and corners between Pb(3)I_6_ and the Pb(1)I_6_ in an adjacent chain 1 (Figure 4). All three PbI_6_ octahedral sites are disordered, and the average bond lengths for Pb(1)-I, Pb(2)-I and Pb(3)-I are 3.200 Å, 3.245 Å and 3.218 Å, respectively. The di-protonated [1,10-phenH_2_]^2+^ cation exhibits a similar type of disorder as observed in **1**, and the water molecules connected to [1,10-phenH_2_]^2+^ are also disordered over the two orientations of the [1,10-phenH_2_]^2+^ cation.

#### 2.3.3. [1,10-phen][Pb_2_I_4_]

Compound **3** crystalizes in the monoclinic space group *I2/a* (No. 15). The 1,10-phenanthroline molecules are attached to Pb atoms via the ring nitrogen atoms and disrupt the original layer structure of the PbI_2_ precursor. The connectivity causes a reduction in the dimensionality of the structure, altering the 2D layer structure (Appendix A) of the starting material into a 1D chain structure (Figure 5). Within the chain are two Pb sites, Pb(1), which is bonded to two N (Pb-N bond length at 2.484 Å) and four I atoms (average Pb-I bond distance is 3.258 Å), and Pb(2), which is bonded to six I atoms (average Pb-I bond distance is 3.242 Å). The Pb(2)I_4_N_2_ edge shares with two other Pb(2)I_6_ to form an infinite zigzag chain. Pb(1) decorates the sides via edge sharing with the phenanthroline groups projecting out into the inter chain space. These chains are oriented in parallel to form layers in the *bc*-plane. These layers stack in an ABAB fashion.

#### 2.3.4. [1,10-phen]_2_[Pb_5_Br_10_]

Compound **4** crystalizes in the monoclinic space group *P2_1_/c* (No. 14) and exhibits a layered structure that contains three different Pb sites (Figure 6). The distorted octahedral Pb(3)Br_6_ cations share trans edges to form infinite chains along the *c*-direction. The distorted octahedral Pb(2) cations share corners with the Pb(3) chains in a staggered fashion to connect the chains into a sheet structure in the bc-plane. This arrangement leaves cavities in the sheets. The seven coordinated Pb(1) cations share edges to form Pb_2_Br_8_N_4_ dimers, where the four nitrogen atoms originate on two trans 1,10-phenanthroline units. These dimers fill the cavities, connecting to the sheet via edge sharing and protrude out on opposite sides. The 1,10-phenanthroline units attached to the Pb_2_Br_8_N_4_ dimers project into the interlayer space and interdigitate with those from adjacent layers. Select bond length and bond angle values are given in Appendix A. Unlike the layered structure of **4**, the structure of the starting reagent, PbBr_2,_ is a 3D structure (Appendix A). The presence of 1,10-phenanthroline results in a dimensional reduction by separating the structure into layers.

#### 2.3.5. [1,10-phenH][SbI_4_]·(H_2_O)

Compound **5** crystallizes in the triclinic space group *P*$\bar{1}$ (No. 2), adopting a one-dimensional (1D) structure that consists of infinite SbI_2_I_4/2_ chains (Figure 7). The octahedral SbI_6_ groups share trans edges to generate infinite zigzag chains. The [1,10-phenH] groups are not connected to the chains but rather are located between groups of four chains. The SbI_2_I_4/2_ chains are negatively charged, requiring a positive charge on each [1,10-phenH] group. In addition, each [1,10-phenH]^+^ group has a water molecule hydrogen bonded to it (Appendix A). Similar to **1** and **3**, the [1,10-phenH]^+^/H_2_O complex is disordered over two positions by a two-fold rotational axis. The SbI_2_I_4/2_ infinite chains are oriented along the *x*-axis and have an average Sb-I bond length of 3.049 Å. The SbI_3_ precursor crystallizes in trigonal space group *R*$\bar{3}$ (No. 148) and exhibits a layered structure. Similar to **3** and **4**, the introduction of 1,10-phenanthroline leads to a dimensional reduction from a 2D layered to a 1D chain structure, although, as opposed to in **3** and **4**, the [1,10-phenH]^+^/H_2_O complexes are not directly bonded to the chains.

### 2.4. Infrared Spectroscopy

IR spectroscopy was carried out to study the different environments of 1,10-phenathroline in each material. In the reported structures, we can find three distinct N environments for the 1,10-phenathroline, including singly protonated (phen-H), doubly protonated (phen-H_2_) and bonded to Pb atoms (phen-Pb). In **1** and **5**, the phenanthroline molecule is singly protonated, while in **2**, it is doubly protonated. The existence of an amide N-H bond is confirmed by the appearance of a broad amide N-H stretch around 2500–3000 cm^−1^ (shaded in blue in the spectra for **1, 2** and **5** (Figure 8)) [32,33]. Furthermore, in both **2** and **5**, the presence of a broad O-H stretch around 3200–3500 cm^−1^ (shaded in red in the spectra for **2** and **5** (Figure 8)) is consistent with the presence of a water molecule in the crystal structure [34,35]. An in-depth analysis of vibrational spectra calculations was carried out to understand the shifts in the IR active N-H stretch for protonated phenanthrolines. The calculations indicate that with increasing protonation, the N-H bond becomes stronger as is observed in the experimental spectra (See SI Section 4). For **3** and **4**, the 1,10-phenanthroline molecules are bound to Pb via Pb-N bonds. The FTIR spectra for both compounds exhibit a pattern similar to that found for the pure organic molecule. The peaks observed around 3040–3060 cm^−1^can be attributed to aromatic C-H stretching [34,35].

In the range of aromatic C–H deformations (600–900 cm^−1^), small shifts can be seen for all spectra compared to pure 1,10-phenanthroline (Figure 9). The peak at 852 cm^−1^ in pure 1,10-phenanthroline corresponds to the out-of-plane motions of the H atoms in the center ring and the latter peak at 736 cm^−1^ for H atoms in the heterocyclic rings [32,35,36,37]. The peak at 852 cm^−1^ is red-shifted by ~6 cm^−1^, ~12 cm^−1^ and 24 cm^−1^ for phen-Pb, phen-H and phen-H_2_ compounds, respectively, while the peak at 736 cm^−1^ shifted by 18 cm^−1^ and 24 cm^−1^ for phen-Pb and protonated phenanthroline compounds. These red shifts, along with the reduction of peak intensities compared to free molecules, further indicate that the 1,10-phenanthroline participates in bonding and agrees with the crystal structure data.

### 2.5. Optical Properties

UV-vis diffuse reflectance spectra for powdered samples were collected at ambient temperature. (Figure 10a). The optical absorption edges were measured at 2.87 eV (432 nm), 2.74 eV (452 nm), 3.44 eV (360 nm) and 2.70 eV (459 nm) for **1**, **2**, **4** and **5, respectively**. The bandgap values correspond well with the respective colors of the materials, and for **1**, **2** and **5**, the values lie in the region of visible-light responsive semiconductors. The recorded values agree with reported hybrid iodoplumbate materials in the literature [8,12,13,22,35,38]. In addition, the absorption peak at ~3.91 eV (317 nm) for the reported materials corresponds to the peak at 3.85 eV (322 nm) for pure 1,10-phenanthroline, which may arise due to intraligand π−π^*^ transitions [22,35].

In comparison to the bulk starting materials, blue shifts are observed for **1**, **2** (PbI_2_ = 2.30 eV [39]); **4** (PbBr_2_ = 3.05 eV); and **5** (SbI_3_ = 2.21 eV). This indicates that the dimensional reduction caused by the incorporation of 1,10-phenanthroline leads to minor changes (~0.5 eV) in the bandgaps [11]. Hence, this approach could be used for fine tuning the optical properties of existing metal halides.

Out of the five reported compounds, only compound **4** exhibited fluorescence emission under UV light irradiation. The fluorescence spectra for single crystals of **4** were obtained at an excitation wavelength of 375 nm. The two emission bands at 523 nm and 562 nm may arise due to the two different Pb^2+^ coordination environments that exist in the crystal structure (Figure 10b). Comparison with the starting reagents suggests that material **4** may exhibit photoluminescence (PL) emission due to three scenarios. First, there can be intraligand π−π^*^ transitions in the 1,10-phenanthroline aromatic rings, which results in blue color emission at 400–450 nm in 1,10-phenanthroline. Second, luminescence may result from ligand to metal or metal to ligand charge transfer between the 1,10- phenanthroline ligand and the Pb^2+^ cation. Finally, the orange color emission at 580 nm may be due to ^1^S_0_ ↔ ^1^P_0_ and ^1^S_0_ ↔ ^3^P_J_ transitions of the s^2^ electron of Pb^2+^. Comparing the values for each transition and with similar hybrid materials in the literature, the PL emission of **4** is likely to arise from the inorganic layer rather than from the high energy emission of 1,10-phenanthroline [21,22,38]. However, more experimental data and calculations of electronic structure are needed to confirm the origin of the PL emission.

### 2.6. Bandgap Calculations

Bandgap calculations were carried out for [1,10-phenH][Pb_3.5_I_8_] to gain a better understanding of the electronic structure of the material. The calculated band structure for one orientation of the disordered phenanthroline is shown in Appendix A. There is a significant discrepancy between the calculated band structure and the experimental data (2.87 eV) obtained via diffuse reflectance spectra, which is likely due to the limitations of semi-local DFT functionals that are known to underestimate calculated band gaps [12,13]. Though the band structures fail to accurately predict the band gap for these structures, the calculations do show that the band structure is impacted by the orientation of the phenanthroline molecule.

## 3. Materials and Methods

Denatured ethanol (Beantown Chemicals), lead iodide (99%, Acros Organics), lead bromide (>98%, Acros Organics), antimony iodide (99.90%, STREM Chemicals), 1,10-phenanthroline (99%, Acros), hydroiodic acid (47% aqueous, Beantown Chemicals), hydrobromic acid (33% in glacial acetic acid, Acros Organics) and hydrochloric acid (37% aqueous, VWR chemicals) were purchased from commercial sources and used without further purification.

### 3.1. Syntheses

#### 3.1.1. [1,10-phenH][Pb_3.5_I_8_] and [1,10-phenH_2_][Pb_5_I_12_]·(H_2_O)

PbI_2_ (0.30 mmol, 138 mg) and 1,10-phenanthroline (0.10 mmol, 18 mg) were placed in a 12 mL glass pressure vessel (ACE Glass) with 5.00 mL of distilled water and 5.00 mL of ethanol as the reaction solvent. A total of 1.00 mL of hydroiodic acid was added, which resulted in the formation of a yellow precipitate inside the tube. The glass tube was sealed with a threaded Teflon plug and heated to 160 °C at a rate of 1 °C/min. The temperature was held at 160 °C for 12 h and cooled down at a rate of 1 °C/min to 80 °C where it was held for 6 h. Finally, using the same rate, the temperature was decreased to room temperature. For [1,10-phenH][Pb_3.5_I_8_] (**1**), yellow color needle-shaped crystals were isolated via vacuum filtration. Loosely threading the Teflon plug to allow solvent to escape during the heating cycle results in [1,10-phenH_2_][Pb_5_I_12_]·(H_2_O) (**2**) in addition to **1**. On the other hand, **2** forms as orange plate crystals on the tube wall. Crystals of **1** and **2** were isolated via vacuum filtration, washed with acetone and allowed to air dry. Suitable single crystals were picked for X-ray diffraction and property measurements.

#### 3.1.2. [1,10-phen][Pb_2_I_4_]

PbI_2_ (0.15 mmol, 69 mg) and 1,10-phenanthroline (0.05 mmol, 9 mg) were placed in a 12 mL glass pressure vessel (ACE Glass) with 2.50 mL of distilled water and 2.50 mL of ethanol as the reaction solvent. A volume of 1.00 mL of hydrochloric acid was added, and a pale-yellow color precipitate formed. The glass tube was sealed with a threaded Teflon plug and heated to 150 °C at a rate of 1 °C/min. The temperature was held at 150 °C for 12 h, and the oven was shut off and allowed to cool down to room temperature. A mixture of yellow needles of [1,10-phenH][Pb_3.5_I_8_] (**1**) and yellow plates of [1,10-phen][Pb_2_I_4_] (**3**) was isolated via vacuum filtration. Crystals were washed with acetone and allowed to air dry, and suitable single crystals were picked for X-ray diffraction and property measurements.

#### 3.1.3. [1,10-phen]_2_[Pb_5_Br_10_]

PbBr_2_ (0.03 mmol, 11 mg) and 1,10-phenanthroline (0.01 mmol, 1.8 mg) were placed in a 12 mL glass pressure vessel (ACE Glass) with 5.00 mL of distilled water and 5.00 mL of ethanol as the reaction solvent. A volume of 0.10 mL of hydrobromic acid was pipetted into the mixture, and the glass tube was sealed with a threaded Teflon plug and heated to 160 °C at a rate of 1°C/min. The temperature was held at 160 °C for 12 h and cooled down at a rate of 1°C/min to 80 °C where it was held for 6 h. Finally, using the same rate, the temperature was decreased to room temperature. The reaction results a clear liquid in which pink plate crystals form upon standing overnight at room temperature, when remaining sealed. It is important to note that the crystallinity of the material improves with colder room temperatures of <20 °C. Crystals were isolated via vacuum filtration, washed with acetone and allowed to air dry. Suitable single crystals were picked for X-ray diffraction and property measurements.

#### 3.1.4. [1,10-phenH][SbI_4_]·(H_2_O)

SbI_3_ (0.30 mmol, 150 mg) and 1,10-phenanthroline (0.10 mmol, 18 mg) were placed in a 12 mL glass pressure vessel (ACE Glass) with 5.00 mL of distilled water and 5.00 mL of ethanol as the reaction solvent. A volume of 1.00 mL of hydroiodic acid was added to the mixture, and the glass tube was sealed with a threaded Teflon plug and heated to 160 °C at a rate of 1 °C/min. The temperature was held at 160 °C for 12 h and cooled down at a rate of 1 °C/min to 80 °C where it was held for 6 h. Finally, using the same rate, the temperature was decreased to room temperature. The reaction resulted in yellow plate crystals that were isolated via vacuum filtration. Crystals were washed with acetone and allowed to air dry, and suitable single crystals were picked for X-ray diffraction and property measurements.

### 3.2. Single-Crystal X-ray Diffraction (SXRD)

Single-crystal X-ray diffraction data were collected at 300(2)-303(2) K on a Bruker D8 QUEST diffractometer equipped with an Incoatec IμS 3.0 microfocus radiation source (MoKα, *λ* = 0.71073 Å) and a PHOTON II area detector. The crystals were mounted on a microloop using immersion oil. The raw data reduction and absorption corrections were performed using the Burker APEX3, SAINT+ and SADABS programs [40,41]. Initial structure solutions were obtained with SHELXTL-2017 [42] using direct methods and Olex2 GUI [43]. Full-matrix least-square refinements against *F^2^* were performed with SHELXL software [44]. The crystallographic data and results of the diffraction experiments are summarized in Table 1.

### 3.3. Powder X-ray Diffraction (PXRD)

Powder X-ray diffraction (PXRD) data were collected at room temperature on ground crystalline samples to confirm phase purity (Appendix A). Data were collected on a Bruker D2 PHASER diffractometer (Bruker Corporation, Karlsruhe, Germany) using Cu Kα radiation over a 2θ range 5–40° with a step size of 0.02°.

### 3.4. FTIR Spectroscopy

Vibrational spectra over the range of 4000–650 cm^−1^ were recorded using a PerkinElmer Spectrum 100 FT-IR spectrometer (PerkinElmer Inc., Waltham, MA, USA) equipped with a diamond ATR attachment.

### 3.5. Optical Properties

UV-vis spectra for powdered samples were recorded using a PerkinElmer Lambda 35 scanning spectrophotometer (PerkinElmer Inc., Waltham, MA, USA). The spectrophotometer was operated in the diffuse reflectance mode and was equipped with an integrating sphere. Reflectance data were converted internally to absorbance via the Kubelka–Munk function [45]. Spectra were recorded in the 200–900 nm range.

Photoluminescence data were collected on single crystals using a HORIBA Scientific Standard Microscope Spectroscopy System (HORIBA Scientific, Piscataway, NJ, USA) connected with iHR320 Spectrometer and Synchrony detector operating on Labspec 6 software. Spectra were recorded from 400 to 900 nm using 375 nm laser excitation source, power 0.5 mW, with 10 × UV objective.

### 3.6. Calculation Details

The partial occupancies observed in the crystal create two possible ideal crystal structures, which are shown in Appendix A. Density-functional theory (DFT) calculations were carried out using the projector-augmented wave (PAW) method [46] as implemented in the Quantum ESPRESSO package [47]. Each spin-polarized structural optimization used the PBE exchange-correlation functional [48] with a 60/800 Ry cut-off for the wave-functional density and cold-smearing [49] with a broadening of 0.01 Ry. Van der Waals interactions are critical to obtain accurate structures for these organometallic structures, so all calculations included the XDM dispersion correction [50]. Each optimization calculation also used a 2 × 2 × 6 *k*-point mesh and had a total energy convergence criterion of 10^−6^ Ry and force convergence of 10^−5^ Ry/Å. Following structural optimization, band structures were calculated for both structures with a 4 × 4 × 12 *k*-point mesh for energy and 302 *k*-points for the band structure.

## 4. Conclusions

In summary, we report the synthesis, structural and optical characterization of five new halometallate compounds that vary in their structural dimensionality from 1D chains to 2D sheets to 3D structures. It is important to note that minute changes in the solvothermal synthesis parameters cause significant differences in the reaction outcomes with respect to crystal structures and the dimensionality of the materials. Three different coordination environments are observed for the 1,10-phenanthroline ligand, un-protonated 1,10-phen, singly protonated [1,10-phenH]^+^ and doubly protonated [1,10-phenH_2_]^2+^, which is confirmed by the appearance of the N-H stretch and peak shifts in the range of aromatic C-H deformations in the FTIR data. The optical properties of phase pure samples indicate that the bandgap values lie in the region of visible-light responsive semiconductors for **1**, **2** and **5**.

## Figures and Tables

**Figure 1 molecules-27-00280-f001:**
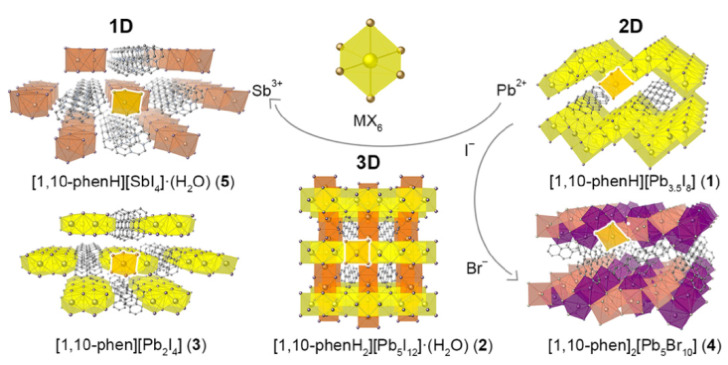
The structures of compounds [1,10-phenH][Pb_3.5_I_8_] (**1**), [1,10-phenH_2_][Pb_5_I_12_]·(H_2_O) (**2**), [1,10-phen][Pb_2_I_4_] (**3**), [1,10-phen]_2_[Pb_5_Br_10_] (**4**) and [1,10-phenH][SbI_4_]·(H_2_O) (**5**) categorized according to structural dimensionality.

**Figure 2 molecules-27-00280-f002:**
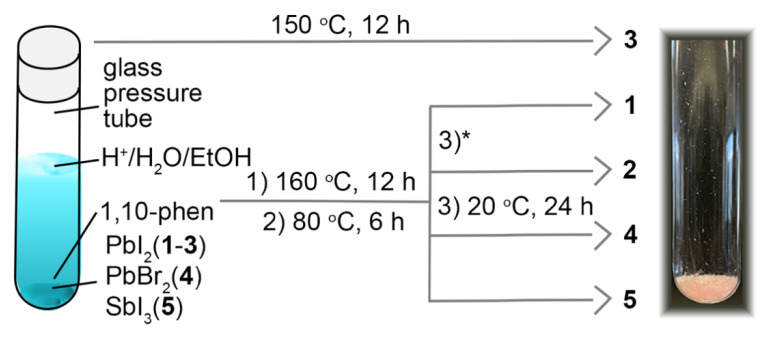
Schematic diagram of the reaction conditions used in synthesis of **1**–**5**. * For material **2**, all reaction conditions were the same as material **1**, and it was produced only when the solvent was evaporated during synthesis.

**Figure 3 molecules-27-00280-f003:**
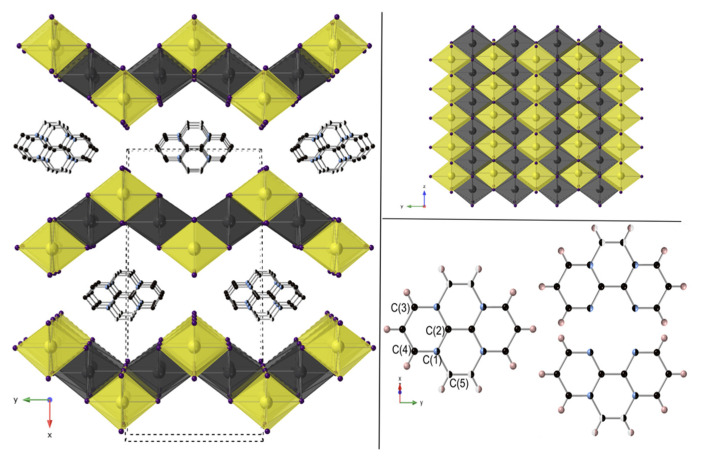
View of [1,10-phenH][Pb_3.5_I_8_] down the *c* axis; unit cell shown in black dashed lines (**left**). View of single [Pb_3.5_I_8_]^−^ layer down the *a* axis (**top right**) and the disordered 1,10-phenanthroline molecule and its two orientations (**bottom right**). Pb(1) is shown in partially filled dark gray color spheres. Pb(2), I, C, N and H shown in yellow, purple, black, blue and pink spheres, respectively.

**Figure 4 molecules-27-00280-f004:**
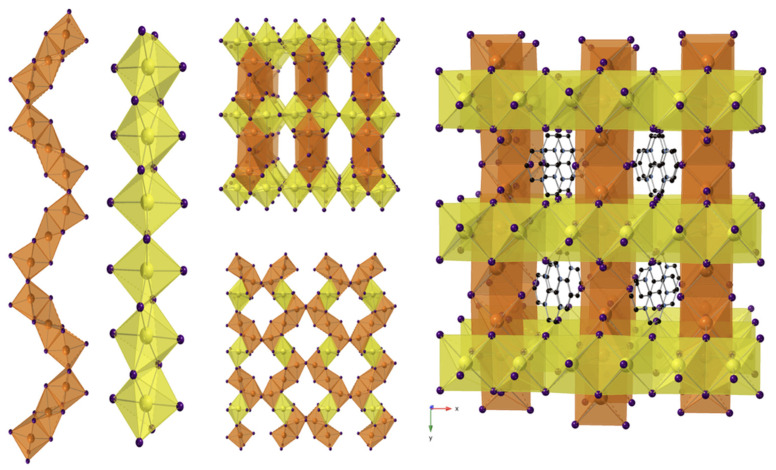
View of chain 1 (orange), view of chain 2 (yellow), the top view from *b*-axis (**top middle**), the side view from a-axis (**bottom middle**) and the view from c-axis of [1,10-phenH_2_][Pb_5_I_12_]·(H_2_O) 3D structure (**left**). Pb(1) and Pb(2) shown in orange octahedra, Pb(3) in yellow octahedra; I and C atoms are shown in purple and black spheres, respectively. H and O atoms are not shown for clarity.

**Figure 5 molecules-27-00280-f005:**
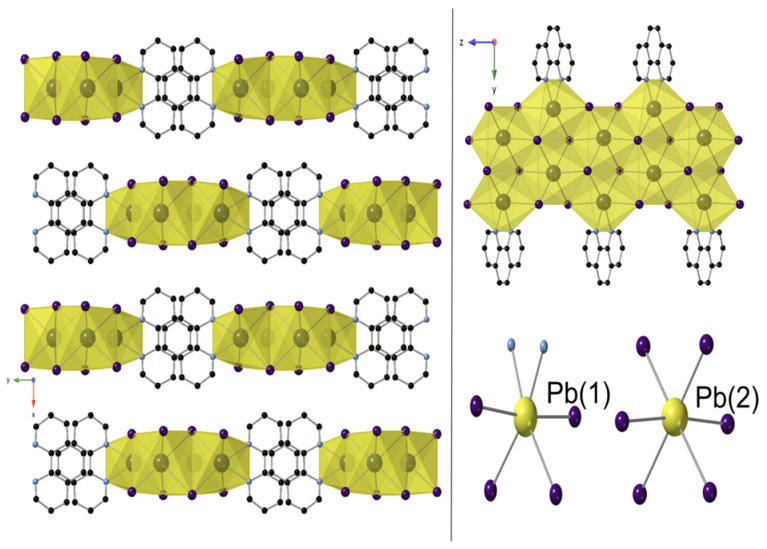
View of [1,10-phen][Pb_2_I_4_] down the c-axis (**left**). View of one 2D chain of [1,10-phen][Pb_2_I_4_] down the a-axis (**top right**). The two Pb sites (**bottom right**). Pb in yellow octahedra; I, C and N shown in purple, black and blue spheres, respectively. H atoms are not shown for clarity.

**Figure 6 molecules-27-00280-f006:**
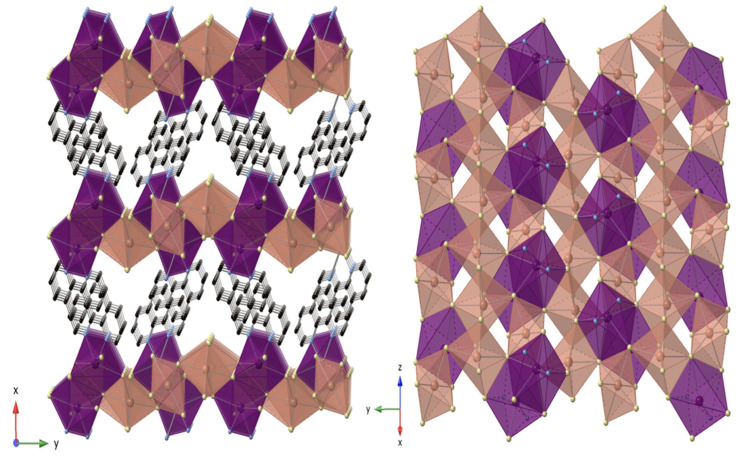
View of [1,10-phen]_2_[Pb_5_Br_10_] down the *c*-axis (**left**). View of single layer (**right**). Pb1 shown in purple polyhedra, Pb2 and Pb3 in beige octahedra. Br, C and N are shown in yellow, black and blue spheres. H atoms are not shown for clarity.

**Figure 7 molecules-27-00280-f007:**
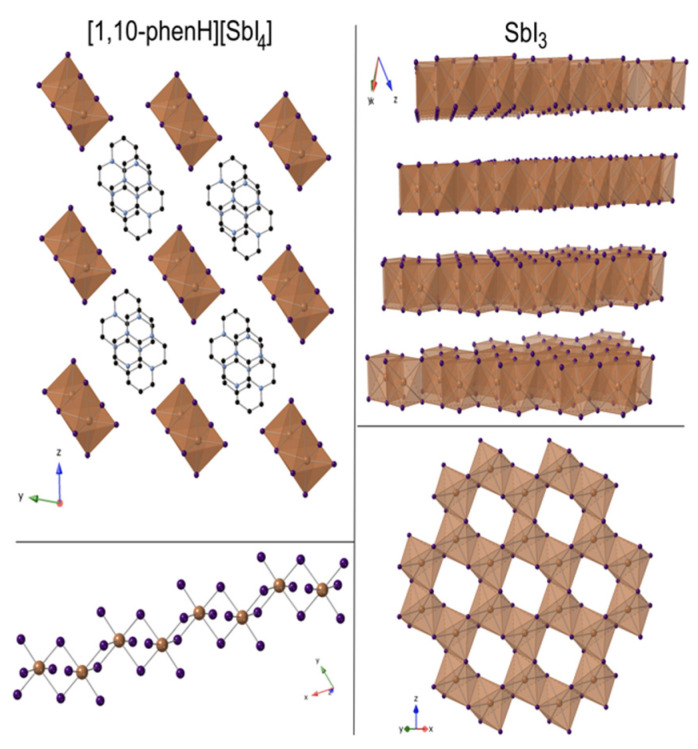
View of [1,10-phenH][SbI_4_]·(H_2_O) down the *a*-axis (**top left**); SbI_4_ chains running along *a*-axis (**bottom left**). View of SbI_3_ layered structure (**top right**) and a single layer of SbI_3_ (**bottom right**). Sb, I, C and N are shown in beige, purple, black and blue spheres. H and O atoms are not shown for clarity.

**Figure 8 molecules-27-00280-f008:**
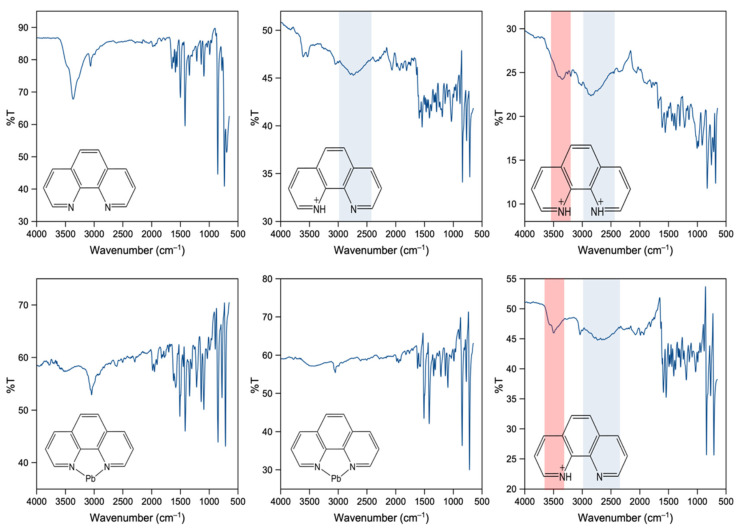
FTIR spectra for pure 1,10-phenanthroline (**top left**), **1** (**top middle**), **2** (**top right**), **3** (**bottom left**), **4** (**bottom middle**) and **5** (**bottom right**).

**Figure 9 molecules-27-00280-f009:**
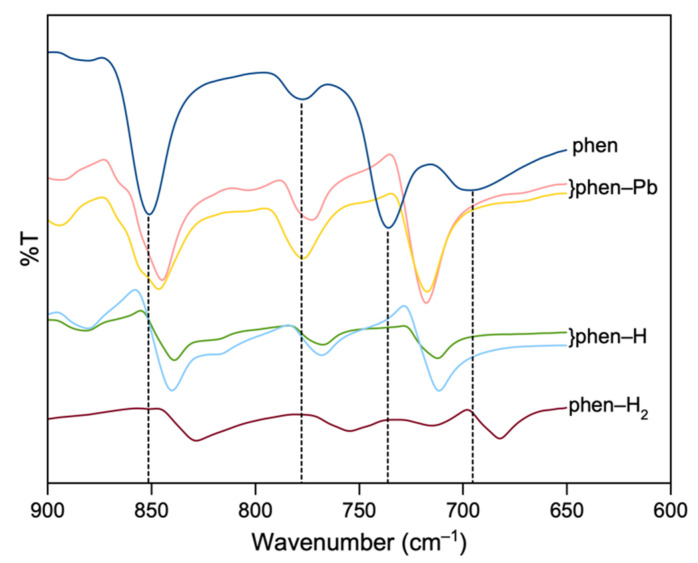
FTIR spectra for 1,10-phenanthroline (dark blue), **1** (green), **2** (brown), **3** (yellow), **4** (pink) and **5** (light blue) in the range 600–900 cm^−1^.

**Figure 10 molecules-27-00280-f010:**
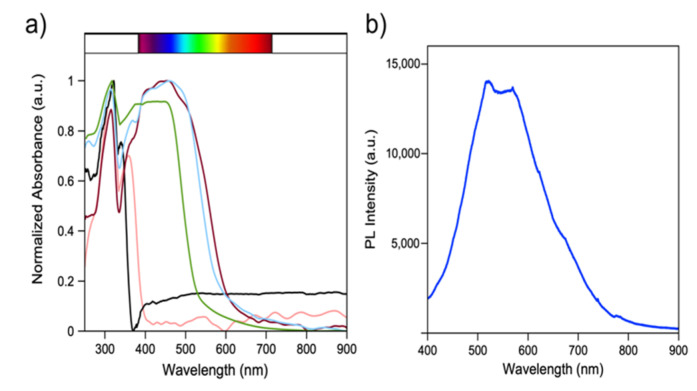
(**a**) UV-vis spectra for 1,10-phenanthroline (black), **1** (green), **2** (brown), **4** (pink) and **5** (light blue). (**b**) The emission spectrum of [1,10-phen]_2_[Pb_5_Br_10_] obtained at an excitation wavelength of 375 nm.

**Table 1 molecules-27-00280-t001:** The crystallographic data for the reported materials **1**–**5.**

Chemical Formula	[C_12_H_9_N_2_][Pb_3.5_I_8_] (1)	[C_12_H_10_N_2_][Pb_5_I_12_]·(H_2_O) (2)	[C_12_H_8_N_2_][Pb_2_I_4_] (3)	[C_12_H_8_N_2_]_2_[Pb_5_Br_10_] (4)	[C_12_H_9_N_2_][SbI_4_]·(H_2_O) (5)
Formula weight	2024.16	2756.97	1102.18	2195.46	828.58
Crystal system	Monoclinic	Orthorhombic	Monoclinic	Monoclinic	Triclinic
Space group	*C2/m*	*Cmme*	*I2/a*	*P2_1_/c*	$P\bar{1}$
a, Å	26.2810(3)	17.9551(3)	15.6100(5)	13.2524(14)	7.7387(9)
b, Å	13.0845(18)	21.0222(3)	16.0694(5)	17.8894(15)	11.1522(13)
c, Å	4.4842(6)	10.7977(2)	7.9333(3)	8.4643(9)	12.0271(13)
α, deg.	90	90	90	90	77.477(4)
β, deg.	97.137(5)	90	95.425(1)	107.942(4)	79.148(4)
γ, deg.	90	90	90	90	70.754(4)
V, Å^3^	1530.1(4)	4075.65(12)	1981.10(12)	1909.1(3)	948.94(19)
ρ_calcd_, g/cm^3^	4.394	4.493	3.695	3.819	2.900
Radiation (λ, Å)	MoKα, 0.71073	MoKα, 0.71073	MoKα, 0.71073	MoKα, 0.71073	MoKα, 0.71073
µ, mm^–1^	30.014	29.686	23.200	32.462	7.958
T, K	301(2)	299(2)	299(2)	302(2)	301(2)
Crystal dim., mm^3^	0.05 × 0.04 × 0.02	0.07 × 0.06 × 0.05	0.05 × 0.02 × 0.01	0.07 × 0.03 × 0.01	0.06 × 0.03 × 0.02
2*θ* range, deg.	2.81–31.77	4.81–56.58	5.07–59.19	5.55–52.74	4.80–59.95
Reflections collected	27,193	131,166	30,501	58,421	62,776
Data/restraints/parameters	1414/0/67	2684/0/69	2782/1/93	3906/0/196	5519/7/200
*R_int_*	0.0524	0.0491	0.0601	0.0800	0.0340
Goodness of fit	1.120	1.181	1.057	1.175	1.065
R_1_(I > 2σ(I))	0.0427	0.0164	0.0443	0.0422	0.0204
wR_2_ (all data)	0.1160	0.0303	0.1275	0.0851	0.0388
Largest diff. peak/hole, e∙Å−3	3.80/−1.43	0.84/−0.66	1.39/−1.90	1.34/−1.24	0.56/−0.67

## Data Availability

CCDC 2109543-2109546, 2110112 contain the supplementary crystallographic data for this paper. These data can be obtained free of charge via www.ccdc.cam.ac.uk/data_request/cif, by emailing data_request@ccdc.cam.ac.uk or by contacting The Cambridge Crystallographic Data Centre, 12 Union Road, Cambridge CB2 1EZ, UK; Fax: +44-1223-336033.

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
