# Peer review of "Investigation of Metastable Low Dimensional Halometallates"

_molecules, 2022, doi:10.3390/molecules27010280_

Round 1

Reviewer 1 Report

The authors synthesized and characterized five new halometallates, namely [1,10-phenH][Pb3.5I8] (1), [1,10-phenH2][Pb5I12]·(H2O) (2), [1,10-phen][Pb2I4] (3), [1,10-phen]2[Pb5Br10] (4) and [1,10-phenH][SbI4]·(H2O) (5), which exhibit rich structural and dimensional diversities. Although this work looks solid, there are still many problems that need to be solved. Therefore, I would like to recommend its publication in Molecules after a major revision.

Some detailed comments are listed below:

  • Since the authors focus the topic of metastabilities of halometallates, the discussions in this aspect are missing. For example, water stability or thermal stability? Some related comments and experiments should be added.
  • For crystal structures:
  • The cif files should be provided.
  • I have some doubts for the molecular compositions of compounds 2 and 5. After all, IR spectra indicated that there are no obvious stretching vibration peaks of H2O molecules (3400-3500 cm-1). In addition, for 2, why the molecular formula is [1,10-phenH2][Pb5I12]·(H2O) instead of [1,10-phenH][Pb5I12]·(H3O); For 5, why the molecular formula is [1,10-phenH][SbI4]·(H2O) instead of [1,10-phen][SbI4]·(H3O). Moreover, the ligands in the acidic conditions may be decomposed into ammonium ions. In other words, the current characterizations are not enough.
  • To understand their compositions more accurately, EA, TGA and TG-MS, especially the latter, may be required.
  • Some comments about the disorders and restrictions in the structures should be given.
  • Their asymmetric unit figures are needed.
  • In page 1, line 16, the compound 4 with the band gap of 3.44 eV cannot be a candidate for visible-light responsive semiconductor applications. Please revise.
  • The calculated band structure looks very strange. As shown in Fig. 10b, there is almost no optical band gap value, indicating the inaccuracy of current calculated method. In addition to the precision, the discussions in the manuscript should also be revised.
  • The UV-vis spectrum of 3 is missing, why?
  • More syntheses details (e.g., yield, color, treatment) of 1-5 should be added. In addition, it is wonderful if the author could summarize the syntheses by a schematic diagram.
  • PXRD patterns with the higher angles, such as 5-75°, are required. In addition, the PXRD data of 2 may be recollected due to some impurities peaks at about 7°.
  • In page 14, line 417, are the IR spectra obtained by calculations? Are the calculated results consistent with the experimental ones?
  • There exist many mistakes in the references, such as refs. 3-6, 7, 9, 12-13, 19-23, 27-30 and 36-37, please check.
  • There are also some grammar, spelling and format mistakes in the manuscript, such as:
  • In page 1, lines 10-12;
  • Tables 1; please check.

Author Response

Reviewer #1

The authors synthesized and characterized five new halometallates, namely [1,10-phenH][Pb3.5I8] (1), [1,10-phenH2][Pb5I12]·(H2O) (2), [1,10-phen][Pb2I4] (3), [1,10-phen]2[Pb5Br10] (4) and [1,10-phenH][SbI4]·(H2O) (5), which exhibit rich structural and dimensional diversities. Although this work looks solid, there are still many problems that need to be solved. Therefore, I would like to recommend its publication in Molecules after a major revision.

Author response: We thank the reviewer for the kind review and comments. We have addressed the issues and revised the paper as requested.

Some detailed comments are listed below:

  • Since the authors focus the topic of metastabilities of halometallates, the discussions in this aspect are missing. For example, water stability or thermal stability? Some related comments and experiments should be added.

Author response: We agree with the reviewer and have added a paragraph to the beginning of the results and discussion section to introduce our thoughts on metastability. Paragraph pasted below.

There are many ways to think about metastable materials and exactly what that term implies about the nature of a material, its composition, its structure, and its stability. One way is to consider metastable materials as structures and compositions that form in a local energy minimum and that are not the most thermodynamically stable phase that exists at the global energy minimum.  Synthesizing such metastable materials therefore requires reaction conditions that kinetically result in the formation of a metastable phase and that are mild enough, often by being carried out at low temperatures, to avoid sliding into the global energy minimum. In this context, metastable- and kinetic-phase, are often used interchangeably.  Generally, solid state reactions, carried out at high temperatures, result in the most thermodynamically stable phases and, therefore, are not typically used to target metastable structures and compositions.  That is better done via low temperature solution routes, such as low melting fluxes, or mild hydrothermal conditions, that operate under conditions in which a metastable material will persist to be isolated, rather than thermally convert to a more stable phase.  The halometallate materials discussed in this paper were all synthesized under extremely mild conditions where numerous local minima appear to exist side by side and where slight perturbations of the synthetic conditions resulted in the stabilization and subsequent isolation of one or another structure and composition.

  • For crystal structures:
  • The cif files should be provided.

Author response: The CIF files are accessible through the ICSD accession codes mentioned CCDC 2109543-2109546, 2110112  It was not specifically asked for during the manuscript submission and we assumed that the above codes, which make the cif files available free of charge, would be sufficient.

  • I have some doubts for the molecular compositions of compounds 2 and 5. After all, IR spectra indicated that there are no obvious stretching vibration peaks of H2O molecules (3400-3500 cm-1).

Author response: We believe the peak around 3200 – 3500 cm-1 in both 2 and 5 corresponds to the OH stretch of water molecules and it is consistent for hydrated phenanthroline compounds in literature (ref 25, 26). We have shaded the peaks in red color for clarity in the new figure (Figure 8).

  • In addition, for 2, why the molecular formula is [1,10-phenH2][Pb5I12]·(H2O) instead of [1,10-phenH][Pb5I12]·(H3O); For 5, why the molecular formula is [1,10-phenH][SbI4]·(H2O) instead of [1,10-phen][SbI4]·(H3O). Moreover, the ligands in the acidic conditions may be decomposed into ammonium ions. In other words, the current characterizations are not enough.

Author response: It is a good question and one that using SXRD we cannot answer.  Using SXRD we are unable to confidently fix the position for the H+ proton due to its low electron density. Therefore, we fix the proton to the most basic atom present and theorize that the N is protonated. In IR characterization, we observe a broad peak at 2500 – 3000 cm-1 which confirms the presence of protonated N. We added a discussion of this issue in the crystal structure determination section.

  • To understand their compositions more accurately, EA, TGA and TG-MS, especially the latter, may be required.

Author response: We do not have access to TG-MS and therefore cannot perform this measurement. The compositions are based on the single crystal structures and, other than the issue of the proton location, not in doubt.  Considering that we have to pick crystals for several phases, the likelihood of a small impurity throwing off the EA is significant.  The phase we were confident that it was 99% phase pure we sent out for EA.  The results for (PhenH)SbI4•H2O came out as C 17.48 (17.39 calc), H 1.32 (1.34 Calc), N 3.35 (3.38 Calc). We added this to the SI.

  • Some comments about the disorders and restrictions in the structures should be given.

Author response: The disorders and restrictions in the structures were stated in the crystal structure solutions section that was previously included in SI. We have now moved this section, which provides great detail about the structure determination, into the text under Section 2.2 in the revised paper.

  • Their asymmetric unit figures are needed.

Author response: Unit cell Figures are now attached to the SI. (Figure S1-S6).

  • In page 1, line 16, the compound 4 with the band gap of 3.44 eV cannot be a candidate for visible-light responsive semiconductor applications. Please revise.

Author response: We agree with the reviewer and have changed the text accordingly.

  • The calculated band structure looks very strange. As shown in Fig. 10b, there is almost no optical band gap value, indicating the inaccuracy of current calculated method. In addition to the precision, the discussions in the manuscript should also be revised.

Author response: We curtailed the discussion in the paper and moved the figure into the SI.  In the SI we added a more detailed explanation of the shortcoming of DFT. 

  • The UV-vis spectrum of 3 is missing, why?

Author response: We were unable to produce phase pure sample for material 3. Therefore, the UV-vis spectrum was not collected. We were able to isolate few crystals only for SXRD and IR characterization.

  • More syntheses details (e.g., yield, color, treatment) of 1-5 should be added. In addition, it is wonderful if the author could summarize the syntheses by a schematic diagram.

Author response: We agree with the reviewer and have added a schematic diagram for the reaction scheme (new Figure 2).

  • PXRD patterns with the higher angles, such as 5-75°, are required. In addition, the PXRD data of 2 may be recollected due to some impurities peaks at about 7°.

Author response: We represented the PXRD patterns with 5–40° for clarity as we felt that putting in the 5-65 degree range makes it hard to see the details.  We have now also added PXRD patterns with the higher angles (5–65°) to the SI (Figure S9-S16). The impurity peaks for material 2 are due to the presence of trace amounts of material 1. We have revised the PXRD pattern for material 2.

  • In page 14, line 417, are the IR spectra obtained by calculations? Are the calculated results consistent with the experimental ones?

Author response: We calculated the IR spectra for 1,10-phenanthroline molecule, [1,10-phenH]+ and [1,10-phenH2]2+ cations to compare the peak shifts in IR active N–H stretch. The calculated data are consistent with the experimental data and are discussed in detail under Section 4 in the SI.

  • There exist many mistakes in the references, such as refs. 3-6, 7, 9, 12-13, 19-23, 27-30 and 36-37, please check.

Author response: We apologize for the mistakes in the references as our reference manager software had a mix-up with journal names. We have corrected all the references and thank the reviewer for pointing out these mistakes.

  • There are also some grammar, spelling and format mistakes in the manuscript, such as:
  • In page 1, lines 10-12;

Author response: We have corrected the grammar and spelling mistakes.

  • Tables 1; please check.

Author response: The table formats have changed, when uploaded to the journal. We have revised the table accordingly.

Reviewer 2 Report

The article written by Hans-Conrad zur Loye and coworkers reports five perovskites that vary from 1D to 3D crystal structures. The metal organic hybrid materials are synthesized using various iodoplumbates with 1,10-phenantroline that depending on the crystallization conditions (pH, Temperature and reaction mixture) the outcomes changed considerably.  A thorough structural analysis using X-ray crystallography has been carried out including the substitution of Pb2+ cation for Sb3+.  The photophysical properties of the perovskites has been also studied in detail which showed that the bandgap values fall in the region of visible-light responsive semiconductors for crystals 1, 2, 4 and 5.

The article is important in the field of hybrid metal-organic materials, particularly in the area of perovskites as it gives new insights on the crystallization methods showing that small variations on the initial conditions can lead towards completely different topologies.  Structure-function relationship has been addressed in detail in the current work focusing on the photophysical properties with potential applications as semiconductors.

The article can be accepted after very minor revisions.

i) It is written that crystal structure 2 crystallizes in the Cmme space group. This is wrong and should be fixed in the text and in Table 1. In the supporting information Cmme should be changed too.

ii) In Table 1, only one structure shows the temperature at which the data was recorded. The authors should indicate the temperature.

iii) In section 3.3 (Powder X-day diffraction).  Was the PXRD data recorded at room temperature? This should be indicated too because the experimental data is compared against the simulated from SCXRD.

iv) The authors mention in the title and very briefly at the introduction that the reported materials are metastable. In my opinion, this is very important aspect that maybe needs to be emphasized a little bit more in the introduction. Why these materials are metastable?

Author Response

Reviewer #2

The article written by Hans-Conrad zur Loye and coworkers reports five perovskites that vary from 1D to 3D crystal structures. The metal organic hybrid materials are synthesized using various iodoplumbates with 1,10-phenantroline that depending on the crystallization conditions (pH, Temperature and reaction mixture) the outcomes changed considerably.  A thorough structural analysis using X-ray crystallography has been carried out including the substitution of Pb2+ cation for Sb3+.  The photophysical properties of the perovskites has been also studied in detail which showed that the bandgap values fall in the region of visible-light responsive semiconductors for crystals 1, 2, 4 and 5.

The article is important in the field of hybrid metal-organic materials, particularly in the area of perovskites as it gives new insights on the crystallization methods showing that small variations on the initial conditions can lead towards completely different topologies.  Structure-function relationship has been addressed in detail in the current work focusing on the photophysical properties with potential applications as semiconductors.

The article can be accepted after very minor revisions.

We thank the reviewer for the kind review and comments on our comprehensive analysis. We have addressed all the issues raised by the reviewer.

  1. i) It is written that crystal structure 2 crystallizes in theCmmespace group. This is wrong and should be fixed in the text and in Table 1. In the supporting information Cmme should be changed too.

Author response: The IUCR has updated space group 67 to be Cmme.  The e represents that there is a double glide plane (aka two translational directions) perpendicular to the c-direction and not just an a glide plane.

  1. ii) In Table 1, only one structure shows the temperature at which the data was recorded. The authors should indicate the temperature.

Author response: In previous table the temperature range for all compounds were mentioned in a merged row that was lost when the journal formatted the paper prior to sending it out to the reviewers. We have now added the individual temperatures for all compositions in separate columns for clarity.

iii) In section 3.3 (Powder X-day diffraction).  Was the PXRD data recorded at room temperature? This should be indicated too because the experimental data is compared against the simulated from SCXRD.

Author response: It was recorded at room temperature and that is now mentioned in the text.

  1. iv) The authors mention in the title and very briefly at the introduction that the reported materials are metastable. In my opinion, this is very important aspect that maybe needs to be emphasized a little bit more in the introduction. Why these materials are metastable?

Author response: Author response: We agree with the reviewer and have added a paragraph to the beginning of the results and discussion section to introduce our thoughts on metastability.

There are many ways to think about metastable materials and exactly what that term implies about the nature of a material, its composition, its structure, and its stability. One way is to consider metastable materials as structures and compositions that form in a local energy minimum and that are not the most thermodynamically stable phase that exists at the global energy minimum.  Synthesizing such metastable materials therefore requires reaction conditions that kinetically result in the formation of a metastable phase and that are mild enough, often by being carried out at low temperatures, to avoid sliding into the global energy minimum. In this context, metastable- and kinetic-phase, are often used interchangeably.  Generally, solid state reactions, carried out at high temperatures, result in the most thermodynamically stable phases and, therefore, are not typically used to target metastable structures and compositions.  That is better done via low temperature solution routes, such as low melting fluxes, or mild hydrothermal conditions, that operate under conditions in which a metastable material will persist to be isolated, rather than thermally convert to a more stable phase.  The halometallate materials discussed in this paper were all synthesized under extremely mild conditions where numerous local minima appear to exist side by side and where slight perturbations of the synthetic conditions resulted in the stabilization and subsequent isolation of one or another structure and composition.

Round 2

Reviewer 1 Report

Dear,

I would like to recommend its publication in Molecules after a minor revision. The following issue had better be resolved.

1) Some related references should be cited about the new comments of metastability.

2) Although the authors made some comments for the molecular compositions. For compounds 2 and 5, I still have some doubts after all the authors also admitted the presence of the protonated N.  In other words, ammonium ion is possible. Therefore, it is wonderful to provide their TGA results.

Author Response

Reviewer #1

I would like to recommend its publication in Molecules after a minor revision. The following issue had better be resolved.

Author response: We thank the reviewer for the kind review and comments. We have addressed the issues and revised the paper as requested.

  • Some related references should be cited about the new comments of metastability.

Author response: We have added several references (new references 23-31) that generally address and or define the concept of metastability.

2) Although the authors made some comments for the molecular compositions. For compounds 2 and 5, I still have some doubts after all the authors also admitted the presence of the protonated N.  In other words, ammonium ion is possible. Therefore, it is wonderful to provide their TGA results.

Author response: We now include the TGA plot for (phenH)Pb3.5I8, one of the protonated compounds, in the SI of the manuscript.  As you can see, the sample completely decomposed in a single step and the weight loss approached 100%.  It is not possible to extract information concerning the possible presence of ammonia from these data. Since our samples all contain lead and clearly volatilize during TGA measurements, we stopped to measure the materials by TGA once we realized this.

We are also still confused why the reviewer is concerned about ammonium ions.  There was no ammonia used in the synthesis and we could find no evidence in the literature that supports the conversion of phenanthroline to ammonia.  If there is a reference to this, we would appreciate receiving it.
